# Identifying depression subtypes and investigating their consistency and transitions in a 1-year cohort analysis

Carolin Oetzmann[1]*, Nicholas Cummins[1], Femke Lamers[2,3], Faith Matcham[4], Sara Siddi[5], Katie M. White[1], Josep Maria Haro[5], Srinivasan Vairavan[6], Brenda W. J. H. Penninx[2,3], Vaibhav A. Narayan[7], Matthew Hotopf[1], Ewan Carr[1]

1 King's College London-Institute of Psychiatry, Psychology & Neuroscience, London, United Kingdom, 2 Department of Psychiatry, Amsterdam UMC, Amsterdam, The Netherlands, 3 Amsterdam Public Health, Mental Health Program, Amsterdam, The Netherlands, 4 School of Psychology, University of Sussex, Falmer, United Kingdom, 5 Parc Sanitari Sant Joan de Déu, Fundació Sant Joan de Déu, CIBERSAM, Universitat de Barcelona, Barcelona, Spain, 6 Janssen Research and Development, LLC, Titusville, NJ, United States of America, 7 Davos Alzheimer's Collaborative, Geneva, Switzerland

* carolin.1.oetzmann@kcl.ac.uk

**Data Availability Statement:** All code and data are available at: https://github.com/carolinoetz/MPlus-Code-for-IDS-SR-Subtyping.

## Abstract

Major depressive disorder (MDD) is defined by an array of symptoms that make it challenging to understand the condition at a population level. Subtyping offers a way to unpick this phenotypic diversity for improved disorder characterisation. We aimed to identify depression subtypes longitudinally using the Inventory of Depressive Symptomatology: Self-Report (IDS-SR). A secondary analysis of a two-year cohort study called Remote Assessment of Disease and Relapse in Major Depressive Disorder (RADAR-MDD), which collected data every three months from patients with a history of recurrent MDD in the United Kingdom, the Netherlands, and Spain (N = 619). We used latent class and latent transition analysis to identify subtypes at baseline, determined their consistency at 6- and 12-month follow-ups, and examined transitions over time. We identified a 4-class solution: (1) severe with appetite decrease, (2) severe with appetite increase, (3) moderate severity and (4) low severity. These same classes were identified at 6- and 12-month follow-ups, and participants tended to remain in the same class over time. We found no statistically significant differences between the two severe subtypes regarding baseline clinical and sociodemographic characteristics. Our findings emphasize severity differences over symptom types, suggesting that current subtyping methods provide insights akin to existing severity measures. When examining transitions, participants were most likely to remain in their respective classes over 1-year, indicating chronicity rather than oscillations in depression severity. Future work recommendations are made.

## Introduction

Depressive disorders affect approximately 5% of adults and are a major cause of disability worldwide [1, 2]. Major depressive disorder (MDD) is defined by persistent low mood and/or

**Funding:** C.O. is funded by the UK Medical Research Council (MR/N013700/1) and King's College London member of the MRC Doctoral Training Partnership in Biomedical Sciences. The RADAR-CNS project has received funding from the Innovative Medicines Initiative 2 Joint Undertaking under grant agreement No 115902. This Joint Undertaking receives support from the European Union's Horizon 2020 research and innovation programme and EFPIA (www.imi.europa.eu). This communication reflects the views of the RADAR-CNS consortium and neither IMI nor the European Union and EFPIA are liable for any use that may be made of the information contained herein. This paper represents an independent research part funded by the National Institute for Health Research (NIHR) Biomedical Research Centre at South London and Maudsley NHS Foundation Trust and King's College London. The views expressed are those of the authors and not necessarily those of the NHS, the NIHR or the Department of Health and Social Care. The funders had no role in study design, data collection and analysis, decision to publish, or preparation of the manuscript.

**Competing interests:** M.H. is the principal investigator of the RADAR-CNS programme, a precompetitive public–private partnership funded by the Innovative Medicines Initiative and the European Federation of Pharmaceutical Industries and Associations. The programme received support from Janssen, Biogen, MSD, UCB and Lundbeck. S.V is an employee of Janssen Research & Development, LLC and hold company stocks/stock options. All other authors declare no competing interests. This does not alter our adherence to PLOS ONE policies on sharing data and materials.

anhedonia combined with various other symptoms [3]. Depression is an incredibly heterogeneous condition. According to the Diagnostic Statistical Manual 5 (DSM-5) criteria, patients can theoretically present with 1,497 different symptom combinations and receive the same MDD diagnosis [3, 4]. This heterogeneity, combined with frequent comorbid conditions, presents a barrier to understanding the mechanisms underlying depression and identifying new treatments. Identifying depression subtypes using symptom-based approaches has a long history [5]. Recent attempts have aimed at identifying new phenotypic classes or verifying classes defined by the DSM-5 [6–9].

Past studies have labelled depressive subtypes to reflect diagnostic criteria, such as atypical or melancholic depression from the DSM. However, systematic reviews aggregating data at an item level have found no consistent evidence for MDD subtypes [10, 11]. While inconsistent results likely reflect study heterogeneity (i.e., differences in statistical methods, model fit criteria, symptom assessment, and coding) and clinical severity [11], these reviews could also suggest that existing methodological approaches are unable to detect subtypes.

Past reviews make three key recommendations to address inconsistencies in subtyping research. First, van Loo et al. [11] call for improved statistical methods, such as techniques to account for classification error or methods to test the assumption of local independence (i.e., that the membership in the latent class entirely explains correlations between the observed class indicators) [12]. Due to the bidirectional nature of many depression symptoms (appetite or sleep increase/decrease, retardation/hyperactivation), violations of this assumption are common and can lead to differences being observed between classes that are due to methodological artefacts [13]. This is critical as even small amounts of unmodeled dependence can bias class prevalence and indicator probabilities [12]. Second, Ulbricht et al. [10] stress the need for increased transparency in modelling approaches to facilitate better comparisons of models in review studies. Third, Ulbricht et al. [10] also call for improved generalisability of subtyping studies by using real-world samples of depressed patients representative of different clinical states rather than relying on recruitment from psychiatric clinics.

The present study addressed these limitations in three ways. First, we used statistical techniques that account for uncertainty in class assignments (such as three-step latent transition analysis), offer reduced bias when incorporating auxiliary variables, and allow testing of local independence assumptions at each time point [14, 15]. Second, we transparently published all analytical code to facilitate replication. Third, we used a real-world sample that included individuals in remission and relapse at baseline and followed them during their naturalistic course. By addressing these limitations, we aim to extend past phenotypic subtyping studies in MDD. Our specific objectives were:

1. To investigate phenotypic subtypes at baseline in a sample of people with MDD;

2. To determine if the same baseline subtypes are identified at 6- and 12-month follow-ups in the same sample; and

3. To examine how participants move between subtypes over time.

## Methods

### Study population and design

The Remote Assessment of Disease and Relapse–Major Depressive Disorder (RADAR-MDD) study was a prospective observational cohort study that aimed to monitor the course of illness and predict relapse in individuals with recurrent MDD using remote measurement technologies. The study enrolled 623 participants who (1) met DSM-5 diagnostic criteria for non-

psychotic MDD within the past 2 years and (2) had experienced at least two recurrent episodes of depression in their lifetime. Individuals were excluded from participating if they had a lifetime history of bipolar disorder, dementia, psychosis or MDD with psychotic features, a history of moderate to severe drug or alcohol dependence in the last 6 months, or a history of major medical disease, that might impact their ability to participate in normal daily life [16].

Participants were followed for up to 24 months, completing validated outcome assessments of depression every 3 months. The study was conducted at three sites: King's College London (United Kingdom), Amsterdam University Medical Centre (The Netherlands), and Centro de Investigación Biomédica en Red (Spain). Participants were recruited from diverse sources, including volunteer registers of people with MDD and clinical samples of people attending mental healthcare services. The recruitment period began on 30th November 2017 and ended on 3rd June 2020. The RADAR-MDD protocol [16] and cohort profile and retention [17] have been published elsewhere.

The study obtained ethical approval from the Camberwell St Giles Research Ethics Committee (REC reference: 17/LO/1154) in London, from the CEIC Fundacio Sant Joan de Deu (CI: PIC-128–17) in Barcelona, and from the Medische Ethische Toetsingscommissie VUmc (METc VUmc registratienummer: 2018.012 –NL63557.029.17) in the Netherlands. All participants provided written informed consent.

## Measures

Depression symptom severity was measured using the Inventory of Depressive Symptomatology–Self Report (IDS-SR; [18]) at the Dutch and Spanish sites the validated translated versions of the IDS-SR were used. While the full IDS-SR contains 28 items, for this analysis, we focused on 14 items representing the 9 core symptoms of MDD [3]. These were: low mood, anhedonia, weight decrease, appetite decrease, weight increase, appetite increase, insomnia, hypersomnia, psychomotor retardation, psychomotor agitation, fatigue, diminished concentration, thoughts of suicide or death, and guilt. We focused on the core symptoms only (1) to mirror past subtyping studies that used the same domains (e.g., [8, 10]); (2) to avoid sparseness in our analysis, given the small sample size available for latent transition models [19]; (3) to minimise local independence violations caused by repeated examination of overlapping domains; and (4) to mirror items used in the well-validated shorter IDS scale: Quick Inventory of Depressive Symptomatology (QIDS; [20]). A more practical consideration is that fewer items would make it easier to implement a screening tool in future.

For this analysis, a score of 2 or 3 for each symptom was coded as "present"; and 0 or 1 score as "not present". Finally, whereas the IDS-SR includes three items assessing insomnia (initial insomnia, middle insomnia and early morning awakening), we combined these into a single score (by taking the median of the three items) to ensure insomnia did not overly influence the identified subtypes.

## Statistical analyses

Our analysis was in three parts, reflecting our three primary objectives. First, we used latent class analysis (LCA) to identify symptom subtypes separately for the baseline, 6-month, and 12-month assessments. LCA derives categorical latent variables to identify clusters of individuals exhibiting similar patterns of categorical indicators [19]. LCA categorises individuals by taking a person-centred approach that identifies patterns of associations among variables [21]. Second, we used latent transition analysis (LTA) to assess the measurement invariance of the classes identified at each timepoint. Thirdly, we used a three-step LTA model to estimate transitions between subtypes over time. In a supplementary analysis, we also considered predictors

of class membership, as we felt it necessary to investigate how individuals in the various classes differed in terms of baseline characteristics.

**Objective 1: Identifying subtypes at individual timepoints.**   We conducted a cross-sectional LCA analysis at baseline and 6- and 12-month follow-ups. We identified the optimal number of classes by estimating models with increasing numbers of classes (from two to six) and assessing model fit based on clinical interpretability and statistical fit indices (Akaike Information Criterion (AIC; [22]); Bayesian Information Criterion (BIC; [23]); sample size adjusted BIC (aBIC; [24]), for which a lower value indicates better fit, and entropy for which a higher value indicates better fit [25]. Higher importance was given to the aBIC as this value has been shown to be a more reliable indicator of model fit for categorical latent class models compared to other information criteria [26].

We tested for violations of the local independence assumption using bivariate residual associations (BVR). These identify potentially problematic residual associations between class indicators by comparing the observed counts to the expected counts (calculated under the assumption of local independence) [12]. While there are no clear rules for what is considered a high BVR, Asparouhov and Muthén [27] suggest that residual pairs with a Pearson test statistic >30 represent severe violations. We adopted a conservative approach by treating all pairs with a test statistic >15 as violations. After detecting a violation, the residual correlation was included in the model, resulting in improved model fit and a higher quality measurement model [12].

**Objective 2: Determine if the same subtypes are identified at each timepoint.**   To assess measurement invariance across timepoints, we estimated two 'one-step' LTA models combining all timepoints (baseline, 6-month, and 12-month). We considered two models: a restricted model (item response probabilities are held equal across timepoints) and an unrestricted model (item response probabilities are freely estimated). We compared model fit between these two models according to fit indicators (AIC, BIC, and aBIC) and a likelihood-ratio difference test [19]. We then repeated this procedure with higher temporal resolution (using measurements at baseline, 3, 6, 9, and 12 months) to assess if the assumption of measurement invariance holds when considering more timepoints.

**Objective 3: Examine how participants move between classes over time.**   We used a three-step LTA model to consider transitions between classes over time. Compared to the one-step model (above), the three-step LTA provides a more robust test of transitions and enables a more confident integration of covariates and predictors into the model, as it reduces bias by accounting for uncertainty in class assignment (i.e., measurement error) [14, 15].

Following Asparouhov and Muthén [14], we conducted a three-step LTA with measurement invariance (see Appendix K-N in [14] for details of the procedures). For Step 1, we estimated a joint LCA model with measurement invariance. Here, the three LCA models (corresponding to the three timepoints) are estimated in parallel but independently, holding all thresholds equal to obtain a longitudinal model with measurement invariance. For Step 2, we estimated a separate LCA at each timepoint, holding all parameters to their values estimated in Step 1. From Step 2, for each timepoint, we extracted the 'most likely class' indicator for each participant and the measurement error for each latent class variable. For Step 3, we combined the indicators into a single LTA model where class membership was estimated from the 'most likely class' indicators with prefixed error rates obtained in Step 2.

**Supplementary analysis: Predictors of class membership.**   For participants with complete information at baseline, 6 months and 12 months, we conducted a secondary analysis to determine if there were clinical or demographic differences between the identified classes. This involved adding predictors of class membership to the three-step LTA model estimated above (Objective 3). Odds ratios were estimated using multinomial logistic regression, where

the outcome was the four-category latent variable representing class membership. Each predictor was tested separately. We chose potential predictors based on previous subtyping studies, data availability, and clinical perspectives: gender [7, 28], age [9], family history of depression (i.e., yes or no having a parent, child, or sibling with diagnosed MDD) [29], physical health comorbidity and/or mental health comorbidity [28, 29], cardiometabolic health comorbidity (i.e., diabetes, high blood pressure, stroke, heart trouble) [6, 30], and presence of high-levels of lifetime trauma [28], as assessed by scoring >6 on the List of Threatening Experiences Questionnaire (LTE-Q; [31]). We additionally considered the use (i.e., yes/no) of specific medications due to their likely influence on depressive symptoms. For example, antidepressants [32], specifically Mirtazapine [33], and antipsychotics [34] have been linked with weight gain; selective serotonin reuptake inhibitor (SSRI) and serotonin-norepinephrine reuptake inhibitors (SNRI) with nausea and weight or appetite loss [35, 36].

Each model was adjusted for age and gender, except for models testing age and gender, which were estimated without adjustment. All P-values were adjusted for multiple comparisons. The null hypothesis was that the presence of a covariate did not contribute significantly to the classification of participants. The significance level was $P < 0.05$ [19].

All LTA and LCA analyses were conducted using Mplus version 8.8 [37]. Corrections for multiple comparisons were made using the Benjamin-Hochberg method [38] in R [39]. Missing data were assumed to be missing at random and thus handled by full information maximum likelihood estimation (FIML), which uses all available information in the observed data to estimate model parameters. All models were estimated with at least 500 random sets of starting values to reduce the chance of suboptimal solutions. The Mplus code for the analyses is available at https://github.com/carolinoetz/MPlus-Code-for-IDS-SR-Subtyping.

## Results

A visual representation of the participant retention rates in the RADAR-MDD sample over the 12-month follow-up is provided in Fig 1. From the total sample (n = 623), we excluded four participants without any IDS-SR assessments at any timepoints, providing an analytical sample of 619 participants at baseline. S1 Table presents the characteristics of the analytical sample. There were 468 (75.6%) females and 151 (24.4%) males, with ages ranging from 18 to 80 (mean: 46.3; standard deviation: 15.2). Most participants identified as 'White British/Dutch' (n = 368, 79.1%), were taking antidepressants at baseline (n = 407, 65.8%) among which SSRIs were the most common (n = 265, 42.8%). Ethnicity data was not collected at the Spanish site.

### Objective 1. Depression subtypes at baseline

We chose a 4-class solution based on model fit, clinical interpretability, and the need to minimise local dependence violations. For base models (Table 1), 4 and 5-class models were

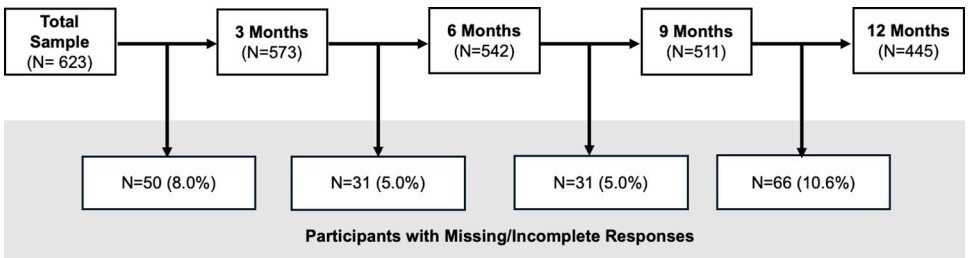

**Fig 1. Participant retention flowchart.**

**Table 1. Model fit and class prevalence for baseline LCA models with varying numbers of classes and residual associations (N = 619).**

| Model fit | 2-Class model | 3-Class model | | 4-Class model | | 5-Class model | 6-Class model |
|---|---|---|---|---|---|---|---|
| | Base model | Base model | Partial dependence model | Base model | Partial dependence model | Base model | Base model |
| AIC | 8141 | 8058 | 8019 | 8007 | 7985 | 7978 | 7968 |
| BIC | 8269 | 8253 | 8219 | 8268 | 8251 | 8305 | 8363 |
| aBIC | 8177 | 8113 | 8076 | 8081 | 8060 | 8070 | 8080 |
| Entropy | 0.84 | 0.80 | 0.77 | 0.75 | 0.74 | 0.74 | 0.77 |
| Class Sizes | | | | | | | |
| Class 1 | 59.1% | 12.2% | 16.4% | 12.3% | 10.4% | 11.3% | 11.8% |
| Class 2 | 40.9% | 33.5% | 34.6% | 12.4% | 14.6% | 14.7% | 1.7% |
| Class 3 | | 54.4% | 49.0% | 36.0% | 35.3% | 27.7% | 11.2% |
| Class 4 | | | | 39.3% | 39.7% | 35.3% | 11.0% |
| Class 5 | | | | | | 11.1% | 28.7% |
| Class 6 | | | | | | | 35.6% |

'Base model' refers to the model without any adjustment for local independence violations. AIC = Akaike information criterion; BIC = Bayesian information criterion.

preferred based on the aBIC. From the 4- and 5-class solutions, we chose the 4-class based on parsimony and slightly improved model fit according to the BIC.

We also chose the 4-class model to minimise local dependence violations present in the 3-class solution that indicated unexplained associations between the items. The 3-class solution presented severe violations of local independence even after modelling initial dependence. Meanwhile, the 4-class model showed only a single minor violation between 'appetite increase' and 'weight increase' (S2 Table). We additionally considered a 'partial dependence' 4-class model where this violation was modelled by accounting for this correlation. Compared to the 'base' 4-class model, the 'partial dependence model' had similar class sizes (Table 1) and almost identical item-response probabilities (S1 Fig). This was expected since dependence violation in the 4-class model was minor (a Pearson statistic of 20.7, whereas previous literature suggests >30 represents a severe violation [27]). Therefore, we opted for the simpler 'base' 4-class solution due to the identical interpretation and comparable class sizes.

Fig 2 shows the 4-class base model's probability of symptom endorsement across the core nine symptoms of depression in the DSM-5. Class 1 (labelled "severe with appetite decrease") was characterised by overall high symptom endorsement compared to the other classes, indicative of a more severe presentation. The class also had the highest probabilities for insomnia, appetite decrease, weight decrease, guilt, psychomotor retardation, and psychomotor agitation compared to other classes. Class 2 ("severe with appetite increase") also presented with high symptom endorsement but was characterised by its highest probabilities for appetite increase, weight increase, and lack of energy. Class 3 ("moderate") overall presented with probabilities lower than Class 1 or 2 but higher than Class 4. Finally, Class 4 ("low") was characterised by the lowest endorsement probabilities compared to the other classes.

See S3 Table for the full table of endorsement probabilities for the 4-class baseline model. For transparency, we also report the probability of symptom endorsement for the 3-class solution in S4 Table.

**Objective 2: Determine if subtypes are consistent across successive timepoints.** When repeating the LCA model at 6- and 12-month timepoints, the 4-class solution again produced optimal fit according to model fit indices (S5 Table), interpretation and parsimony (S2 Fig), and local independence violations (S2 Table). When testing measurement invariance of the three-time point one-step latent transition model, compared to the freely estimated model, model fit was improved for the restricted model (ΔAIC = -105, ΔBIC = -601, ΔAdjusted BIC =

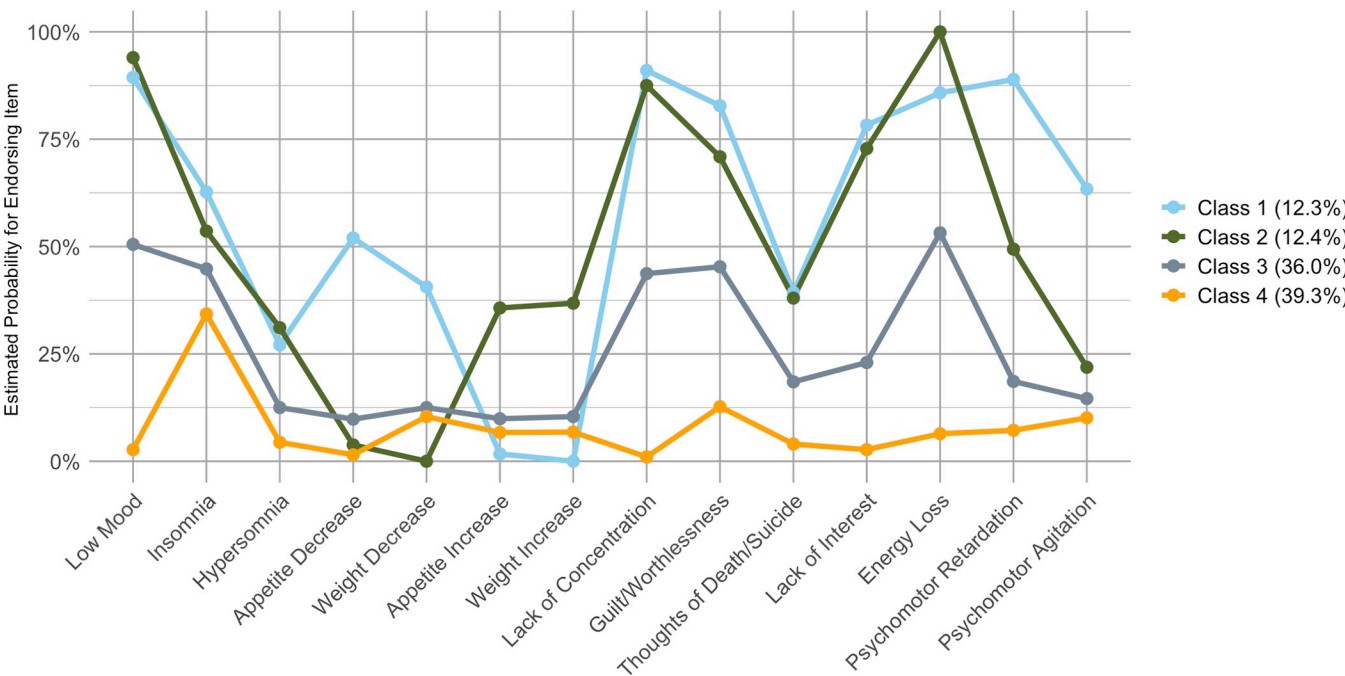

**Fig 2. Probabilities of endorsing depressive symptoms derived from baseline 4-class latent class analysis (N = 619).**

-246), suggesting measurement invariance. This assumption was also formally tested using a nested loglikelihood ratio test ($G^2_2 - G^2_1 = 59.2$, df = 112, *p = 0.999*).

To explore whether the assumption of measurement invariance held when we include more timepoints (S6 Table), we repeated this process for items collected at 0, 3, 6, 9, and 12 months (S6 Table). The assumption for measurement invariance was upheld, suggesting that a consistent 4-class structure was identified at each timepoint.

**Objective 3: Examine transitions between classes over time.** For this analysis, participants with incomplete data at any of the three timepoints (baseline, 6 months and 12 months) were excluded as defined by the three-step procedure, giving an analytical sample of 432. Excluded participants (n = 187) were similar in terms of demographics to the original sample. 146 (78.1%) females and 41 (21.9%) males, with a mean age of 45 (standard deviation: 15.4) ranging from 19 to 75.

Fig 3 presents the class transition probabilities from 0 to 6 and 6 to 12 months based on the three-step LTA. Overall, participants tended to remain in their initial classes at subsequent timepoints (e.g., 61% and 81% of the Class 2: "severe with appetite increase" and Class 4: "low severity" classes, respectively). Of note is the high stability seen between the two more severe classes (Class 1 and Class 2). Here, participants had less than 15% chance of switching between them over time.

See S8 Table for the full results table.

Regarding movements between classes at 0 to 6 months, the largest transition probabilities were seen for moves from Class 4 to Class 3 ("moderate severity") (22%) and from Class 2 to Class 3 (17%). For transitions at 6 to 12 months, the largest probabilities were for Class 3 to Class 4 ("low severity") (25%) and Class 2 to Class 4 (22%).

When looking at the transition probabilities for the one-step 0-, 3-, 6-, 9- and 12-month LTA models, the results appear similar, with participants most likely to remain within the same class over time (see S7 Table).

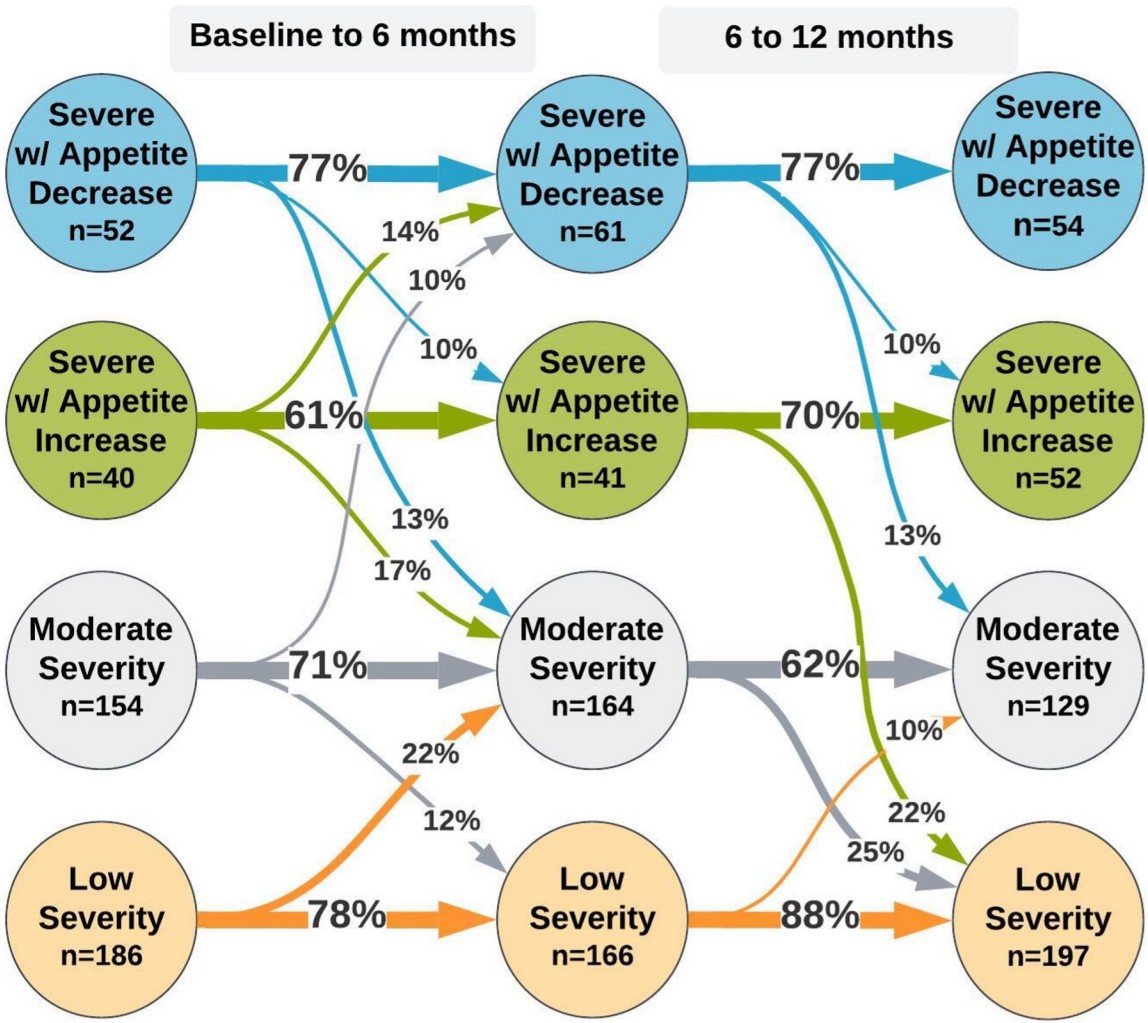

**Fig 3. Transition probabilities and class sizes for three-step LTA model (N = 432).** Probabilities <10% are not shown.

**Supplementary analysis: Predictors of class membership.** We estimated a series of models where each predictor was added (separately, in turn) to the 3-step LTA model. All models were adjusted for age and gender. Here, we focus on the results for Class 1 ("severe with appetite decrease") and Class 2 ("severe with appetite increase") since these classes were separated by symptom type rather than severity and predictors of depression severity are well-established elsewhere [40–43]. Data in S9 Table presents descriptive statistics for predictors across all classes.

We found no evidence of any associations between the chosen predictors and the classification of participants into Class 1 vs. Class 2 (Table 2), suggesting participants in these classes were similar in terms of sex, age, family history, comorbidities, and lifetime trauma. We also found no evidence of differences between these classes in terms of medication use. As would be expected, statistically significant differences were seen across classes of different severity (e.g., between Class 1 and Class 3 or Class 4). For example, we found that participants with a physical comorbidity were more likely to belong to Class 1 ("severe with appetite decrease") than Class 4 ("low"); participants taking SNRIs and those taking antipsychotics were also more likely to belong to Class 1 than Class 4.

**Table 2. Adjusted multinomial logistic regression models of class membership at baseline in relation to participant characteristics (N = 432).**

| Predictor | Class 1 | | Class 2 | | Class 3 | | Class 4 |
|---|---|---|---|---|---|---|---|
| | N = 53 | | N = 30 | | N = 161 | | N = 188 |
| | Severe w/ Appetite Decrease | | Severe w/ Appetite Increase | | *Moderate* | | Low |
| | Ref. | Odds ratio | *Adj. p-value** | Odds ratio | *Adj. p-value** | Odds ratio | *Adj. p-value** |
| Female | 1.0 | 2.1 | 0.55 | 1.0 | 0.99 | 0.9 | 0.94 |
| Age (years/10) | 1.0 | 1.0 | 0.94 | 1.0 | 0.94 | 1.3 | 0.08 |
| Family history of depression | 1.0 | 1.9 | 0.55 | 1.3 | 0.79 | 2.5 | 0.08 |
| Physical health comorbidity | 1.0 | 0.3 | 0.12 | 0.3 | 0.08 | 0.1 | <0.000 |
| Mental health comorbidity | 1.0 | 1.0 | 0.98 | 0.3 | 0.08 | 0.3 | 0.08 |
| Cardiometabolic comorbidity | 1.0 | 0.2 | 0.50 | 0.5 | 0.41 | 0.4 | 0.18 |
| Lifetime traumatic events | 1.0 | 1.9 | 0.55 | 1.1 | 0.94 | 1.1 | 0.94 |
| SSRI | 1.0 | 1.2 | 0.94 | 1.1 | 0.94 | 1.3 | 0.64 |
| SNRI | 1.0 | 0.5 | 0.50 | 0.4 | 0.08 | 0.1 | <0.000 |
| Antipsychotics | 1.0 | 0.2 | 0.21 | 0.3 | 0.08 | 0.2 | <0.000 |
| Mirtazapine | 1.0 | 0.6 | 0.85 | 0.6 | 0.55 | 0.2 | 0.08 |

Ns shown represent class sizes in the model without any covariates. All covariates were entered separately as predictors of baseline class status adjusting for age and gender. Models including only age and gender were estimated independently without any confounders. *P-values were adjusted to account for multiple hypothesis testing using the Benjamin-Hochberg method. All variables except for age are binary. Antidepressant & comorbidity groups are not exclusive. Age: in decades to create more interpretable coefficients. Selective serotonin reuptake inhibitor: SSRI; Serotonin–norepinephrine reuptake inhibitor: SNRI. Lifetime traumatic events, scoring >6 was considered high number of traumatic events. Descriptive results presented in S9 Table.

## Discussion

This study aimed to identify phenotypic subtypes of MDD from self-report symptom assessments, examine their stability over time, and investigate transitions between subtypes. Our analyses consistently identified four distinct subtypes of depression: (1) severe with appetite decrease, (2) severe with appetite increase, (3) moderate severity and (4) low severity. Subtypes mainly differed in terms of severity (the varying likelihood of symptom endorsement) and, for the two more severe classes, the type of neurovegetative symptoms reported. These phenotypic differences in the two more severe classes were defined by high probabilities of endorsing insomnia, appetite decrease, weight decrease, feelings of guilt/worthlessness and psychomotor symptoms for Class 1 (severe with appetite decrease). Meanwhile, Class 2 (severe with appetite increase) was defined by high probabilities of endorsing appetite increase, weight increase and lack of energy symptoms. These differences remained even when accounting for local independence violations. Longitudinally, these same subtypes were observed at different points in time, and participants were most likely to remain in the same class over the one-year follow-up period. We found no evidence of significant differences in sample characteristics between the two severe subtypes. Individuals in Class 1 and 2 were similar in terms of sex, age, family history of depression, experiences of lifetime trauma, comorbidities (physical, mental and/or cardiometabolic) and antidepressant use.

Our findings showed many similarities with previous works. For example, our findings align with Lamers et al. and Rodgers et al. [7, 44] in identifying two more severe subtypes with neurovegetative differences and a more moderate severity subtype. Furthermore, when looking at the individual items which defined their "severe typical" class and "severe atypical" subtypes, similarities can be seen in our work: psychomotor symptoms, appetite/weight decrease and insomnia for Class 1 and appetite/weight increase and leaden paralysis for Class 2. Our findings regarding transitions between subtypes over time were also consistent with previous research, in particular, the finding that participants tended to remain in the same class over the follow-up period [44, 45]. Similarly, Rodgers et al. [7] reported high stability for their severe atypical and moderate classes but lower stability for the severe typical class.

However, our identification of four classes differs from these past studies (e.g., [28]), which have tended to find three classes instead. This divergence could be attributed to differences in study design, as past studies only included participants during a depressive episode, whereas we included people with a history of depression but no/mild current symptoms. The current approach aimed to capture the diverse experiences of depression observed in real-world contexts, as recommended by Ulbricht et al. [10] to improve the generalisability of the findings. Our results also differ regarding predictors of class membership: we identified no significant differences between the severe classes, whereas Rodgers et al. [7] found that those in the severe atypical subtype were more likely to be female and have comorbid eating disorders and psychosis symptoms. Lamers et al. [28] also found a female preponderance, higher body mass index and more frequent metabolic syndrome in the atypical class. Our lack of statistically significant findings could be explained by a lack of statistical power and imprecise measures (e.g., no direct assessment of BMI and metabolic syndrome).

When interpreting our findings, it is important to acknowledge that severity was the primary factor distinguishing our classes rather than differences in phenotypic expression. Where there were phenotypic differences, these were predominantly between the two more severe classes. While this may be expected given our heterogeneous sample, it raises questions about the clinical meaningfulness of these subtypes over and above severity assessments. Van Loo et al. [13] state that subtypes are only meaningful if they provide better clinical insight than current practice (e.g., better treatment matching or furthering understanding of aetiology and pathophysiology). Despite some differences in symptom patterns between our severe subtypes, our identified classes tended to be similar (see Fig 2). Thus, whether relatively minor differences in symptom profiles are sufficient to justify the existence of clinical subtypes beyond existing severity markers remains an open question.

The lack of differences between identified subtypes may reflect inadequate measurement [11]. Whereas we derived subtypes based on the nine core MDD criteria, this overlooks other relevant symptoms and cannot capture the diverse clinical experience of depression. Chevance et al. [46] found that core outcomes important to patients, carers and health professionals, like social and elementary functioning (i.e., self-care, ability to get out of bed), are rarely considered in standard measures of MDD. Biases in retrospective self-report measures, influenced by cognitive heuristics and social desirability, may compromise objectivity [47, 48]. To overcome these limitations, richer and more objective behavioural data collected at higher frequencies, such as digital biomarkers from wearable devices and smartphone apps, could provide a more nuanced understanding of depression symptoms.

## Strengths and limitations

The key strengths of our work lie in replicating and extending the existing depression subtyping literature (e.g., [7, 28]) while addressing several important criticisms of previous studies.

We addressed a need for improved statistical methods by employing a robust three-step LTA approach, surpassing the limitations of the traditional one-step [14, 15]. We conducted comprehensive investigations into violations of local independence at each timepoint, ensuring the integrity of our findings and enabling transparent reporting of our modelling approaches [10, 13].

However, when interpreting our findings, several limitations should be considered. Despite RADAR-MDD being a large study, our sample size was limited. Previous subtyping work has used samples ranging from 61 to 13,424 [10], but little consensus on optimal sample sizes for subtyping analyses exists. Some researchers suggest a minimum sample of 500 for LCA and LTA [49], while simulation studies using the three-step method for LTA have suggested samples of 500 to 2000 [14]. Lower bound sample sizes (like those seen in our 12-month LCA and three-step LTA model) are considered sufficient when paired with high entropy and large transition probabilities [50], as seen in our analysis. Therefore, while our study meets these general criteria, sparseness remains a concern, particularly in the two small severe classes [19]. This particularly impacts the generalisability of regression models to identify predictors of class membership. Furthermore, as participants were most likely to remain in the same class over time, there was insufficient statistical power to assess predictors of transitions between classes.

Furthermore, our findings linking medication use and class membership should be interpreted cautiously since directionality cannot be inferred. Additionally, participants may have been subject to polypharmacy, as such individuals may have been on both appetite-suppressing and appetite-increasing medication at once, notwithstanding the potential impact of non-pharmacological mental health interventions (e.g., exercise) or the effects of other medications on symptoms like weight, sleep, psychomotor symptoms, or concentration, which were not accounted for in our analysis.

## Conclusions

Our study extends the depression subtyping literature by identifying four subtypes: (1) severe with appetite decrease, (2) severe with appetite increase, (3) moderate severity and (4) low severity. These classes were measurement invariant at 6- and 12-month follow-ups, with participants tending to remain in the same class over time. Notably, our findings highlighted differences in severity rather than symptom profile, suggesting that current subtyping approaches may be inadequate for identifying meaningful phenotypic subtypes. More nuanced assessments are needed to provide clinical utility beyond standard severity measures. This study lays the groundwork for future research using robust LTA and LCA methodologies to address limitations noted in the literature. High-frequency, objective data collection assessing a range of symptoms and behavioural domains may provide the necessary nuance to uncover clinically meaningful subtypes. Such subtypes could inform targeted interventions and improve long-term patient outcomes.

## Supporting information

**S1 Table. Demographic and baseline sample characteristics.** The table represents the demographic and clinical sample characteristics at baseline.
(PDF)

**S2 Table. Local independence violations with a person test statistic >15 for the 3 and 4 class LCA models at baseline, 6-months, and 12-months.** The results presented in S2 Table, outline the local dependence violations seen in the 3 and 4-class solutions for baseline, 6-months and 12-months. Baseline results are discussed in the results section. For 6 months

the original 3 and 4 class solutions (i.e., no modelled dependence) presented with 2 and 1 pairs of problematic local dependence between the appetite and weight variables, respectively. In both solutions modelling the dependence removed the concerns. Meanwhile, for the 12-month model violations were only observed in the three-class solution, while the 4-class solution presented no violations.
(PDF)

**S3 Table. Item-response probabilities of endorsing depressive symptoms derived from 4-class baseline latent class analysis (N = 619), no modelled dependence.**
(PDF)

**S4 Table. Probabilities of endorsing depressive symptoms derived from 3-class baseline latent class analysis (N = 619).**
(PDF)

**S5 Table. Model comparisons in cross-sectional latent class analysis at baseline, 6-month and 12-month follow-up.** The results in S5 Table show that the 4-Class solution presented a good middle ground between the 3 and 5-Class solutions across the timepoints. Specifically, at 6-months, 4-class solution presented the best model fit for BIC and featured only a slightly higher aBIC than the 5-class solution. Furthermore, entropy was higher for the 4-class than the 5-class or 6-class solution, as such this was chosen. Similarly at 12-month follow-up, 4-class solution was chosen as it has the best fitting aBIC value, furthermore it has considerably higher entropy than the 6-class solution. The results for baseline are outlined in the results section.
(PDF)

**S6 Table. Probabilities of endorsing depressive symptoms derived from one-step measurement invariant latent transition analysis model: Baseline, 3-, 6-, 9-, 12- month (N = 619).** Here we can see that overall, the derived model shares a similar interpretation to the baseline latent class model and the 6-, 12-month models. It identifies four classes (1) severe with appetite decrease, (2) severe with appetite increase, (3) moderate severity, (4) low severity. Measurement invariance was assessed by comparing model fit in a nested model comparison. Here the restricted model (item response probabilities held equal over time) was preferred over the freely varying model. Specifically, the AIC, BIC and aBIC were lower for the restricted model ($\Delta$AIC = -211.76, $\Delta$BIC = -1203.65, $\Delta$aBIC = -492.493). This assumption was also formally tested using a nested loglikelihood ratio test ($G^2_2 - G^2_1$ = 118.12, df = 224, $p > 0.05$) further suggesting measurement invariance.
(PDF)

**S7 Table. Transition probabilities and latent class sizes for one-step LTA model for baseline, 3-, 6-, 9-, & 12-months (N = 619).** Interpretation of these probabilities suggests that membership in all 4 classes was stable over time and participants were most likely to stay within their same class over time, with probabilities ranging from 94% chance for remaining in class 4 from 9 to 12 months, and 68% chance for remaining in class 3 from baseline to 3 months. Notable between class changes over-time occur for individuals moving from class 2 to class 3. Here those in class 2 at report a 27% chance of moving to class 3 from 3 to 6 months and from 6 to 9 months.
(PDF)

**S8 Table. Transition probabilities and class sizes for three-step LTA model for baseline, 6-months, 12-months.**
(PDF)

**S9 Table. Descriptive crosstabulation of class membership at baseline in relation to participant characteristics.**
(PDF)

**S1 Fig. Item-response probabilities for endorsing depressive symptoms at baseline in 4-class partial dependence model and 4-class base model (no modelled dependence).**
(PDF)

**S2 Fig. Item-response probabilities for endorsing depressive symptoms at baseline, 6-month and 12-month follow-up.** The results presented in S2 Fig, show that across each of the independently run latent class analyses, the item-response probabilities for endorsing a symptom are comparable in the 4-class solutions over time.
(PDF)

## Acknowledgments

We thank our colleagues both within the RADAR-CNS consortium and across all involved institutions for their contribution the recruitment strategy for RADAR-MDD (http://www.radar-cns.org). Furthermore, we would like to thank the FAST-R group, a team with experience of mental health problems and their carers who have been specially trained to advise on research proposals and documentation through the Feasibility and Acceptability Support Team for Researchers (FAST-R): a free, confidential service in England provided by the National Institute for Health Research Maudsley Biomedical Research Centre via King's College London and South London and Maudsley NHS Foundation Trust. We would also like to thank all members of the RADAR-CNS patient advisory board, who all have experience of living with or supporting those who are living with depression, epilepsy or multiple sclerosis.

## Author Contributions

**Conceptualization:** Carolin Oetzmann, Femke Lamers, Faith Matcham, Josep Maria Haro, Srinivasan Vairavan, Brenda W. J. H. Penninx, Vaibhav A. Narayan, Matthew Hotopf, Ewan Carr.

**Data curation:** Carolin Oetzmann, Femke Lamers, Faith Matcham, Sara Siddi, Katie M. White, Brenda W. J. H. Penninx.

**Formal analysis:** Carolin Oetzmann.

**Funding acquisition:** Matthew Hotopf.

**Investigation:** Carolin Oetzmann, Nicholas Cummins, Katie M. White, Brenda W. J. H. Penninx.

**Methodology:** Carolin Oetzmann, Nicholas Cummins, Femke Lamers, Faith Matcham, Sara Siddi, Josep Maria Haro, Srinivasan Vairavan, Brenda W. J. H. Penninx, Vaibhav A. Narayan, Ewan Carr.

**Project administration:** Femke Lamers, Faith Matcham, Sara Siddi, Josep Maria Haro, Vaibhav A. Narayan, Matthew Hotopf.

**Resources:** Carolin Oetzmann.

**Software:** Carolin Oetzmann.

**Supervision:** Nicholas Cummins, Matthew Hotopf, Ewan Carr.

**Visualization:** Carolin Oetzmann.

**Writing – original draft:** Carolin Oetzmann.

**Writing – review & editing:** Carolin Oetzmann, Nicholas Cummins, Femke Lamers, Faith Matcham, Sara Siddi, Josep Maria Haro, Srinivasan Vairavan, Brenda W. J. H. Penninx, Vaibhav A. Narayan, Matthew Hotopf, Ewan Carr.

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
