## [Decision Letter · Decision Letter 0]

11 Sep 2024

PONE-D-24-23747Identifying Depression Subtypes and Investigating their Consistency and Transitions in a 1-Year Cohort AnalysisPLOS ONE

Dear Dr. Oetzmann,

Thank you for submitting your manuscript to PLOS ONE. After careful consideration, we feel that it has merit but does not fully meet PLOS ONE’s publication criteria as it currently stands. Therefore, we invite you to submit a revised version of the manuscript that addresses the points raised during the review process.

We look forward to receiving your revised manuscript.

Kind regards,

Wajid Mumtaz

Academic Editor

PLOS ONE

Journal Requirements:

“The RADAR-CNS project has received funding from the Innovative Medicines Initiative 2 Joint Undertaking under grant agreement No 115902. This Joint Undertaking receives support from the European Union’s Horizon 2020 research and innovation programme and EFPIA (www.imi.europa.eu). This communication reflects the views of the RADAR-CNS consortium and neither IMI nor the European Union and EFPIA are liable for any use that may be made of the information contained herein. The funding body has not been involved in the design of the study, the collection or analysis of data, or the interpretation of data. This paper represents an independent research part funded by the National Institute for Health Research (NIHR) Biomedical Research Centre at South London and Maudsley NHS Foundation Trust and King’s College London. The views expressed are those of the authors and not necessarily those of the NHS, the NIHR or the Department of Health and Social Care.”

“C.O. is funded by the UK Medical Research Council (MR/N013700/1) and King’s College London member of the MRC Doctoral Training Partnership in Biomedical Sciences. M.H. is the principal investigator of the RADAR-CNS programme, a precompetitive public–private partnership funded by the Innovative Medicines Initiative and the European Federation of Pharmaceutical Industries and Associations. The programme received support from Janssen, Biogen, MSD, UCB and Lundbeck. S.V and QL are employees of Janssen Research & Development, LLC and hold company stocks/stock options. All other authors declare no competing interests.”

5. One of the noted authors is a group or consortium [RADAR-CNS consortium]. In addition to naming the author group, please list the individual authors and affiliations within this group in the acknowledgments section of your manuscript. Please also indicate clearly a lead author for this group along with a contact email address.

6. Please amend your manuscript to include your abstract after the title page.

7.Please review your reference list to ensure that it is complete and correct. If you have cited papers that have been retracted, please include the rationale for doing so in the manuscript text, or remove these references and replace them with relevant current references. Any changes to the reference list should be mentioned in the rebuttal letter that accompanies your revised manuscript. If you need to cite a retracted article, indicate the article’s retracted status in the References list and also include a citation and full reference for the retraction notice.

Reviewers' comments:

Reviewer's Responses to Questions

**Comments to the Author**

1. Is the manuscript technically sound, and do the data support the conclusions?

Reviewer #1: Yes

Reviewer #2: Yes

2. Has the statistical analysis been performed appropriately and rigorously? 

Reviewer #1: Yes

Reviewer #2: Yes

3. Have the authors made all data underlying the findings in their manuscript fully available?

Reviewer #1: Yes

Reviewer #2: Yes

4. Is the manuscript presented in an intelligible fashion and written in standard English?

Reviewer #1: Yes

Reviewer #2: Yes

5. Review Comments to the Author

Reviewer #1: Dear Editors and Authors,

Thank you for the privilege of reviewing the manuscript titled "Identifying Depression Subtypes and Investigating their Consistency and Transitions in a 1-Year Cohort Analysis". In it, the authors describe a clustering analysis of patients with depression over a one year period, using latent class and latent transition analysis to determine subtypes of depression over this period of time.

Strengths

1. Well written introduction that clearly outlines the contribution of this work.

2. Thank you for sharing your code, that is an important step to ensure reproducibility.

3. Good use of a relatively real-world dataset.

4. Great coaching of results with respect to other literature and past work, e.g. the Van Loo et al statement.

5. Very well written and easy to read.

Minor Comments

1. Please mention your specific multiple comparison correction in the text, I found it only in a table caption.

2. There is a missing closing paren on line 38 or 39.

3. Is there a legend for Figure 1, or otherwise somewhere where we can understand which group is which?

I have no Major Comments.

Reviewer #2: Dear Authors,

First, I would like to extend my congratulations on conducting a very interesting study. I have some comments regarding the report.

The study titled “Identifying Depression Subtypes and Investigation their Consistency and Transitions in a 1- Year Cohort Analysis” focuses on, as the title indicates, identifying depression subtypes, as well as examining the consistency and transitions of this subtypes over time. The study identifies four different classes of depression subtypes and demonstrates that these classes remain stable over a one-year period, with participants generally tending to remain within the same class through this time.

Introduction:

Line 29: it’s not mentioned which version of DSM you are referring to.

Method:

I would prefer to see a figure her, where the participant flow is shown, drop-out etc.

Results:

Line 201: figure 1 is not presented in the text. It makes it hard to understand the differences between the different groups. I think it also would be wise to go more in depth in the different symptoms of depression that you are investigating.

In the attachments, figure 1 is missing explanations on the different colors in the figure.

Line 208: Class 3 (“moderate”) overall presented with 208 probabilities lower than Class 1 or 3 but higher than Class 4. I think it is a typing error where class 2 and 3 have been mixed up.

Line 238: Figure 2 is missing.

6. PLOS authors have the option to publish the peer review history of their article (what does this mean?). If published, this will include your full peer review and any attached files.

Reviewer #1: No

Reviewer #2: No

---

## [Author Response · Author response to Decision Letter 0]

5 Oct 2024

Dear Editiors, 

Please see rebuttal letter for all comments regarding the content of the paper. I have also gone ahead and updated the additional sections as per your instructions. Thank you.

---

## [Decision Letter · Decision Letter 1]

13 Nov 2024

Identifying Depression Subtypes and Investigating their Consistency and Transitions in a 1-Year Cohort Analysis

PONE-D-24-23747R1

Dear Dr. Oetzmann,

We’re pleased to inform you that your manuscript has been judged scientifically suitable for publication and will be formally accepted for publication once it meets all outstanding technical requirements.

Kind regards,

Wajid Mumtaz

Academic Editor

PLOS ONE

Additional Editor Comments (optional):

Reviewers' comments:

Reviewer's Responses to Questions

**Comments to the Author**

1. If the authors have adequately addressed your comments raised in a previous round of review and you feel that this manuscript is now acceptable for publication, you may indicate that here to bypass the “Comments to the Author” section, enter your conflict of interest statement in the “Confidential to Editor” section, and submit your "Accept" recommendation.

Reviewer #1: All comments have been addressed

2. Is the manuscript technically sound, and do the data support the conclusions?

Reviewer #1: Yes

3. Has the statistical analysis been performed appropriately and rigorously? 

Reviewer #1: Yes

4. Have the authors made all data underlying the findings in their manuscript fully available?

Reviewer #1: Yes

5. Is the manuscript presented in an intelligible fashion and written in standard English?

Reviewer #1: Yes

6. Review Comments to the Author

Reviewer #1: Thank you for the revisions, I have no further comments and support publication of this manuscript..

7. PLOS authors have the option to publish the peer review history of their article (what does this mean?). If published, this will include your full peer review and any attached files.

Reviewer #1: No

---

## [Editor Report · Acceptance letter]

23 Dec 2024

PONE-D-24-23747R1 

PLOS ONE

Dear Dr. Oetzmann, 

I'm pleased to inform you that your manuscript has been deemed suitable for publication in PLOS ONE. Congratulations! Your manuscript is now being handed over to our production team.

Kind regards, 

on behalf of

Dr. Wajid Mumtaz 

Academic Editor

PLOS ONE